# Acoustic non-Hermitian skin effect from twisted winding topology

Li Zhang[1,2,6], Yihao Yang 📧 [1,2,6✉], Yong Ge[3,6], Yi-Jun Guan[3], Qiaolu Chen[1,2], Qinghui Yan[1,2], Fujia Chen[1,2], Rui Xi[1,2], Yuanzhen Li[1,2], Ding Jia[3], Shou-Qi Yuan 📧 [3], Hong-Xiang Sun 📧 [3✉], Hongsheng Chen 📧 [1,2✉] & Baile Zhang 📧 [4,5✉]

The recently discovered non-Hermitian skin effect (NHSE) manifests the breakdown of current classification of topological phases in energy-nonconservative systems, and necessitates the introduction of non-Hermitian band topology. So far, all NHSE observations are based on one type of non-Hermitian band topology, in which the complex energy spectrum winds along a closed loop. As recently characterized along a synthetic dimension on a photonic platform, non-Hermitian band topology can exhibit almost arbitrary windings in momentum space, but their actual phenomena in real physical systems remain unclear. Here, we report the experimental realization of NHSE in a one-dimensional (1D) non-reciprocal acoustic crystal. With direct acoustic measurement, we demonstrate that a twisted winding, whose topology consists of two oppositely oriented loops in contact rather than a single loop, will dramatically change the NHSE, following previous predictions of unique features such as the bipolar localization and the Bloch point for a Bloch-wave-like extended state. This work reveals previously unnoticed features of NHSE, and provides the observation of physical phenomena originating from complex non-Hermitian winding topology.

[1] Interdisciplinary Center for Quantum Information, State Key Laboratory of Modern Optical Instrumentation, College of Information Science and Electronic Engineering, Zhejiang University, Hangzhou 310027, China. [2] ZJU-Hangzhou Global Science and Technology Innovation Center, Key Lab. of Advanced Micro/Nano Electronic Devices & Smart Systems of Zhejiang, ZJU-UIUC Institute, Zhejiang University, Hangzhou 310027, China. [3] Research Center of Fluid Machinery Engineering and Technology, School of Physics and Electronic Engineering, Jiangsu University, Zhenjiang 212013, China. [4] Division of Physics and Applied Physics, School of Physical and Mathematical Sciences, Nanyang Technological University, 21 Nanyang Link, Singapore 637371, Singapore. [5] Centre for Disruptive Photonic Technologies, The Photonics Institute, Nanyang Technological University, 50 Nanyang Avenue, Singapore 639798, Singapore. [6] These authors contributed equally: Li Zhang, Yihao Yang, Yong Ge. ✉email: yangyihao@zju.edu.cn; jsdxshx@ujs.edu.cn; hansomchen@zju.edu.cn; blzhang@ntu.edu.sg

The revolutionary topological classification of matter (e.g., topological insulators and topological semimetals) is underpinned by the concept of band topology formed by Bloch eigenstates, commonly calculated by the Bloch theorem in an infinitely large system without a boundary[1,2]. Introducing a boundary will only lead to boundary states, but will not change any bulk property, including the band topology, as considered universally true for all Hermitian, or energy-conservative, systems. However, the recently discovered NHSE has shown that non-Hermiticity, originating from loss/gain or non-reciprocity, can cause all the bulk eigenstates to collapse towards the introduced boundary[3–16]. The complete collapse of bulk bands in NHSE has posed a challenge to the bulk-boundary correspondence[3]—a fundamental principle of topological physics—which states that the band topology derived from the bulk dictates the topological phenomena at the boundary.

Understanding the NHSE requires non-Hermitian band topology, a newly introduced set of topological descriptions with features distinct from the Hermitian counterpart[6,17–21]. For example, the nontrivial topology can arise from a non-Hermitian single band, with no symmetry protection, while a Hermitian band topology requires two or more bands, generally under some symmetry protection. So far, all NHSE observations are based on one type of non-Hermitian band topology, in which the complex energy spectrum winds along a closed loop in the complex plane[7–9,15]. The winding direction, characterized by the winding number of $+1$ or $-1$, determines which boundary, the left or right, all bulk eigenstates shall collapse toward in the NHSE[4,6] (see Fig. 1a). In a simplified picture, the collapse direction is consistent with the direction of dominant coupling[22].

Topological windings beyond the simplest single-loop configuration can generate much more complex topology. For example, as recently demonstrated along a frequency synthetic dimension in a photonic resonator, non-Hermitian topological winding can exhibit almost arbitrary topology in momentum space, including a twisted winding that produces two oppositely oriented loops in contact rather than a single loop[23] (see the right panel in Fig. 1b). However, since the characterization assumes an infinitely large system without a boundary, the corresponding actual physical phenomena in a finite system still remain uncharacterized. On the other hand, as predicted by recent theories, the twisted winding may dramatically change the NHSE: the bulk eigenstates can collapse towards two directions, exhibiting the bipolar localization that is inconsistent with the dominant coupling direction; and the contact point between the two loops, named as the Bloch point, corresponds to a Bloch-wave-like extended state, interpolating the left-localized and right-localized eigenstates[22,24] (see the left panel in Fig. 1b). None of these phenomena has been observed.

Here, we experimentally demonstrate the NHSE in a 1D non-reciprocal acoustic crystal exhibiting the twisted non-Hermitian winding topology. Topological acoustics is an ideal platform to explore various topological phenomena, due to the ease with which the properties and couplings of acoustic crystals can be engineered[25,26]. The non-reciprocal coupling, while difficult in other crystal lattices such as photonic crystals, is realized here with directional acoustic amplifiers. With non-reciprocal nearest-neighbor coupling, the crystal exhibits conventional single-winding-type NHSE, similar to previous observations[7–9,15]. When non-reciprocity is applied to the next-nearest-neighbor coupling, the crystal's NHSE transits to the bipolar type that corresponds to the twisted winding topology[22]. A discrete Bloch-wave-like extended mode is also observed at the Bloch point. As the non-reciprocal coupling can be applied between two arbitrary sites, our acoustic platform provides a versatile platform to directly observe physical phenomena arising from more complex unconventional topological windings (see Supplementary Information Note 1).

## Results

**Implementation of acoustic non-Hermitian non-reciprocal coupling**. We start with the design of non-reciprocal coupling. As shown in Fig. 2a, two identical acoustic resonators (labeled "1" and "2") with resonance frequency $\omega_0$ are connected by two narrow waveguides that provide reciprocal coupling $\kappa_1$ (see Supplementary Information Note 2). Note that the two

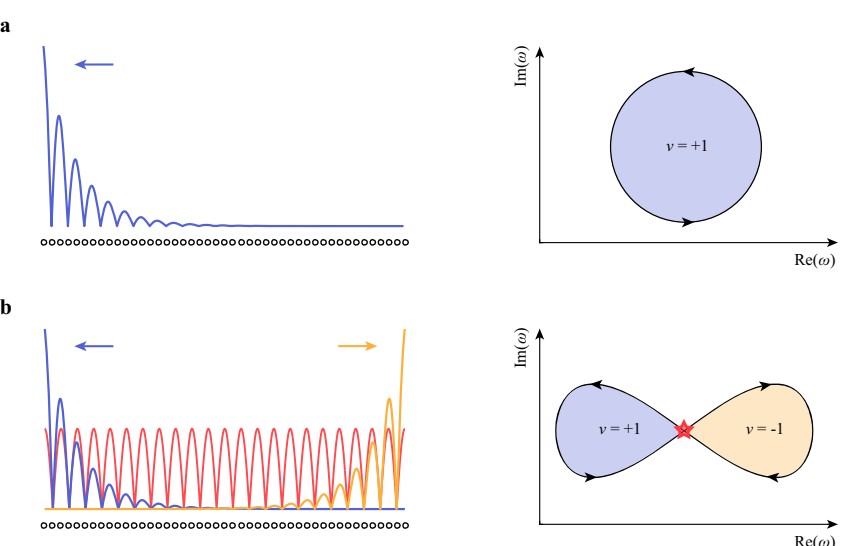

**Fig. 1 Comparison of NHSE arising from different winding topologies. a** In the conventional NHSE, all bulk eigenstates localize at a boundary (left panel). This corresponds to the single-loop winding of the complex energy spectrum in the complex plane (right panel). The winding direction is characterized by the winding number ($v = +1$ in the illustration) that determines which boundary (the left boundary in the illustration) the bulk eigenstates should localize toward. **b** A twisted winding topology consists of two oppositely oriented loops in contact (right panel). The corresponding NHSE exhibits bipolar localization, where bulk eigenstates localize towards two directions (left panel). The contact point (denoted by a red star in the right panel) corresponds to a Bloch-wave-like extended state (red line illustrated in the left panel). In the left panels, the dots at the bottom present the lattice; the arrows indicate the directions that the eigenstates shall collapse toward.

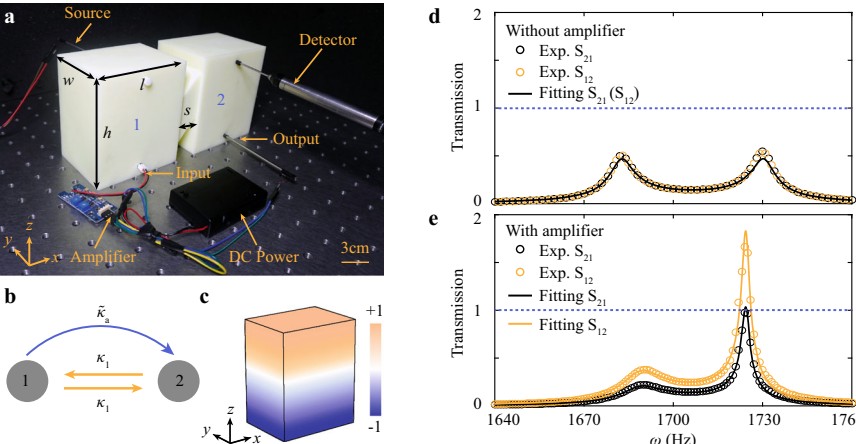

**Fig. 2 Implementation of acoustic non-Hermitian non-reciprocal coupling. a** Photograph of two acoustic resonators (labeled by "1" and "2") connected by two cross-linked narrow waveguides. The non-reciprocal coupling is implemented by an amplifier (equipped with a DC power supply), a microphone (input), and a speaker (output). A source and a detector are used to measure the transmission. **b** Simplified tight-binding model for the setup in (**a**). **c** Simulated acoustic pressure field distribution of the dipole mode in a single resonator. **d**, **e** Experimentally measured (circles) and numerically fitted (solid lines) transmission spectra for samples without and with the amplifier, respectively. The blue dashed line denotes the unity of transmission.

waveguides can enhance the coupling strength. The unidirectional coupling, denoted as $\tilde{\kappa}_a$, is introduced by an additional setup that consists of a directional amplifier (equipped with a direct current (DC) power supply), a speaker (output), and a microphone (input). This additional setup amplifies the sound unidirectionally: the sound is detected at resonator 1, and then coupled to resonator 2 after amplification. This unidirectional amplification introduces non-reciprocity. The tight-binding model (see Fig. 2b) can lead to the following Hamiltonian (see details in Supplementary Information Note 3):

$$H = \begin{bmatrix} \omega_0 - i\gamma_0 & \kappa_1 \\ \kappa_1 + \tilde{\kappa}_a & \omega_0 - i\gamma_0 \end{bmatrix}. \quad (1)$$

Here, $\gamma_0$ arises from the intrinsic loss, including the viscothermal loss. Note that we only consider the dipole mode in a single resonator (see Fig. 2c).

To experimentally retrieve the couplings, we measure both $|S_{12}|$ and $|S_{21}|$ transmission spectra without and with the amplifier (see experimental details in Methods), as shown in Fig. 2d, e, respectively. As expected, when the amplifier is not in operation, i.e., with only the reciprocal coupling, $|S_{12}|$ and $|S_{21}|$ are very close, and there occur two resonance peaks with almost identical amplitude. However, when the amplifier is in operation, we observe three notable features (Fig. 2e). Firstly, $|S_{12}|$ is higher than $|S_{21}|$. This is direct evidence of non-reciprocity. Secondly, the maximum transmission is greater than unity, which comes from gain. Lastly, the two resonance frequencies are different from those in the Hermitian case, indicating that the non-reciprocal coupling has changed the eigenfrequencies of the system. By numerically fitting the measured spectra with the coupled-mode theory (see Supplementary Information Note 4), we can obtain the resonator's resonance frequency $\omega_0 = 1706$ Hz, the reciprocal coupling $\kappa_1 = 24$ Hz, the non-reciprocal coupling $\tilde{\kappa}_a = -11 + 3.9i$ Hz, and $\gamma_0 = 2.13$ Hz, respectively. The phase of $\tilde{\kappa}_a$ results from the phase delay of the amplifier (see Supplementary Information Note 5).

**NHSE from single-winding topology**. We then proceed to demonstrate the NHSE in a 1D crystal that consists of 20 acoustic resonators as shown in Fig. 3a. Here the non-reciprocal coupling is applied between nearest-neighbor resonators, following the

tight-binding model in Fig. 3b (see Methods for more details). The lattice Hamiltonian of the crystal with periodic boundary conditions is $H = \kappa_1 e^{ika} + \kappa_1 e^{-ika} + \tilde{\kappa}_a e^{-ika} + \omega_0 - i\gamma_0$, where $k$ is the wavevector, and the lattice constant $a = 11.6$ cm. The imaginary and real parts of the eigenfrequency spectrum are presented in Fig. 3c. The asymmetric distribution of the imaginary part (yellow curve) with respect to $k = 0$ indicates the non-reciprocity. The corresponding eigenfrequency spectrum in the complex frequency plane is also plotted in Fig. 3d. The spectrum winds around a tilted elliptical loop, along the direction of increasing wavenumber $ka$ from 0 to $2\pi$. For an arbitrary frequency $\omega_b$ in the complex plane (without overlapping with the dispersion band), a quantized integer winding number $\nu$ can be obtained as

$$\nu = \frac{1}{2\pi i} \int_{-\pi}^{\pi} \frac{\partial \omega(k)/\partial k}{\omega(k) - \omega_b} dk \quad (2)$$

where $\omega(k)$ is the eigenfrequency band. Geometrically, the rotation direction and the number of times of this closed-loop enclosing the reference frequency $\omega_b$ determinates the sign and value of $\nu$, respectively[16,18]. Here, owing to the counter-clockwise winding that encloses a finite area containing an arbitrary base frequency $\omega_b$ (e.g., $1700 + 5i$ Hz) for a single time, the winding number is $\nu = +1$, which can also be confirmed numerically by the above equation.

The winding number $\nu = +1$ determines that all eigenstates shall collapse to the left boundary. This also means that, regardless of the location of the source, all waves in the crystal shall propagate only to the left, forming a funneling effect[8]. For the purpose of demonstration, we first launch an acoustic pulse, covering the frequency range from 1680 to 1720 Hz, from site 1 (20), and measure the transmission at site 20 (1). As depicted in Fig. 3e, f, the transmission towards the right boundary almost vanishes, while the transmission towards the left boundary is amplified (also see the measured time-domain results in Supplementary Information Note 6 and Note 7). Such a phenomenon can be used to devise a unidirectional compact waveguide without relying on a large-size bulk to separate the one-way edge states as in Chern insulators[27]. Moreover, we also measure the field distributions in the acoustic crystal when the source is placed at different locations with operating frequency $\omega = 1713$ Hz, as shown in Fig. 3g (see more results at other frequencies in Supplementary Information Note 8). There are two

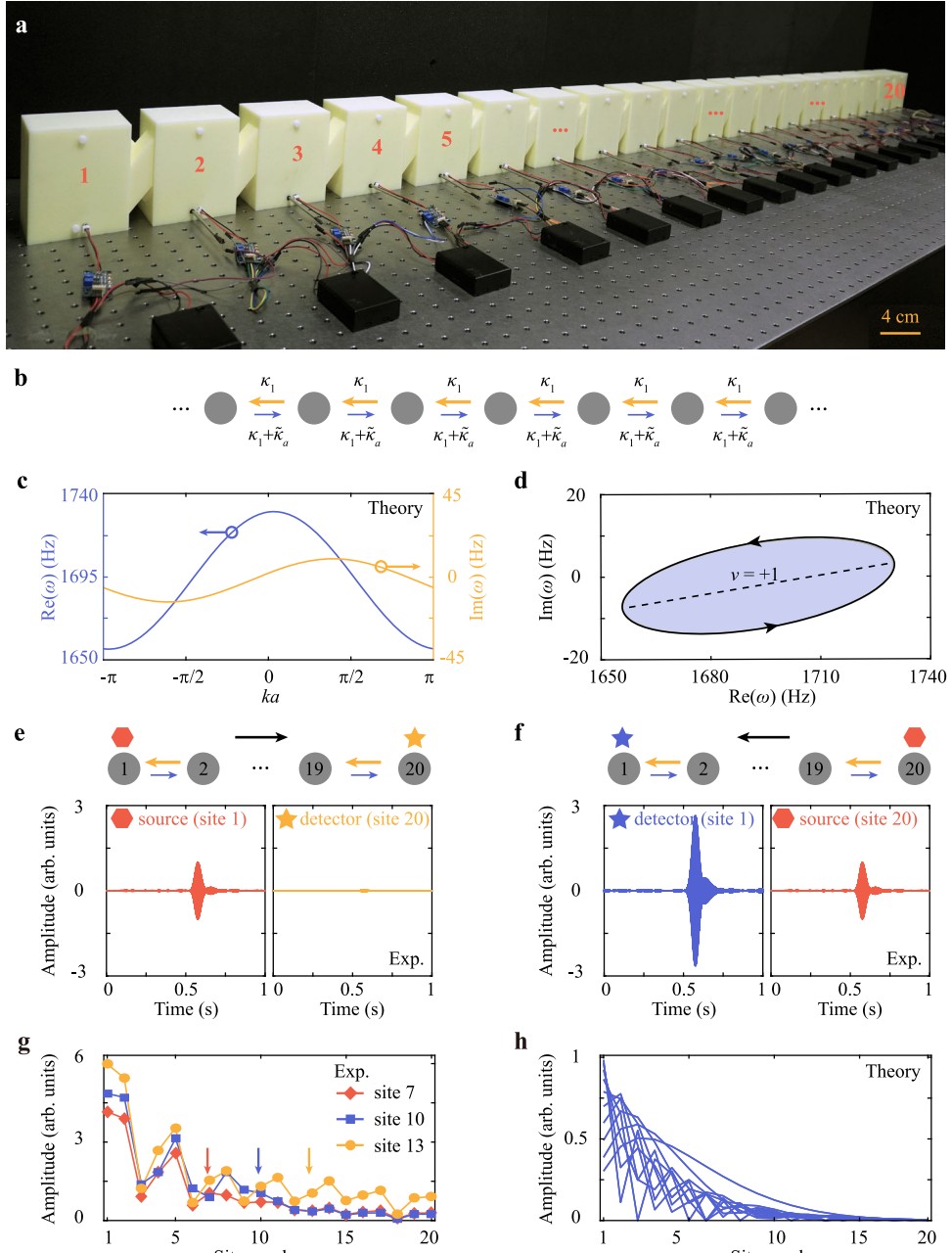

**Fig. 3 Observation of acoustic NHSE from single-winding topology. a** Photograph of the acoustic crystal composed of 20 cross-linked acoustic resonators. Non-reciprocal coupling is applied between adjacent resonators. **b** Tight-binding model with nearest-neighbor non-reciprocal couplings $\kappa_1 + \tilde{\kappa}_a$ and $\kappa_1$, as denoted by the blue and yellow arrows, respectively. **c** Real and imaginary parts of the eigenfrequency spectrum calculated with periodic boundary conditions. **d** The complex eigenfrequency spectrum calculated in (**c**) forms a closed loop in the complex frequency plane, with the winding number $v = +1$. The dashed line corresponds to the eigenfrequency spectrum calculated with the open boundary conditions. **e**, **f** Transmission measurement of a pulse in the time domain. The top schematic diagram indicates the location of the source (hexagon) and the detector (star) along the acoustic crystal. The measured results are normalized by the amplitude at the source location. **g** Measured acoustic pressure field distribution at frequency $\omega = 1713$ Hz when the source is located at sites 7, 10, and 13, respectively. The measured results are normalized by the amplitude at the source location (indicated by little arrows). **h** Superposition of the calculated field distributions of eigenstates in the finite crystal.

striking features of the NHSE. Firstly, the energy is concentered on the left boundary irrespective of the source location. This is different from a conventional Hermitian acoustic crystal, where the energy distribution generally concentrates around the source location (see Supplementary Information Note 9). Secondly, the field distribution localized at the left boundary exhibits stronger amplitude, when the source is placed further away from the left. Those two features are direct evidence of the NHSE[9]. We also numerically calculated the field distribution of eigenstates in the

finite crystal under open boundary conditions, which further corroborates the localization of all eigenstates at the left boundary (see Fig. 3h).

**NHSE from twisted-winding topology.** A remarkable advantage of our acoustic crystal is that the non-reciprocity can be conveniently applied not only to the nearest-neighbor coupling but also to the long-range coupling, the latter of which is the key element of

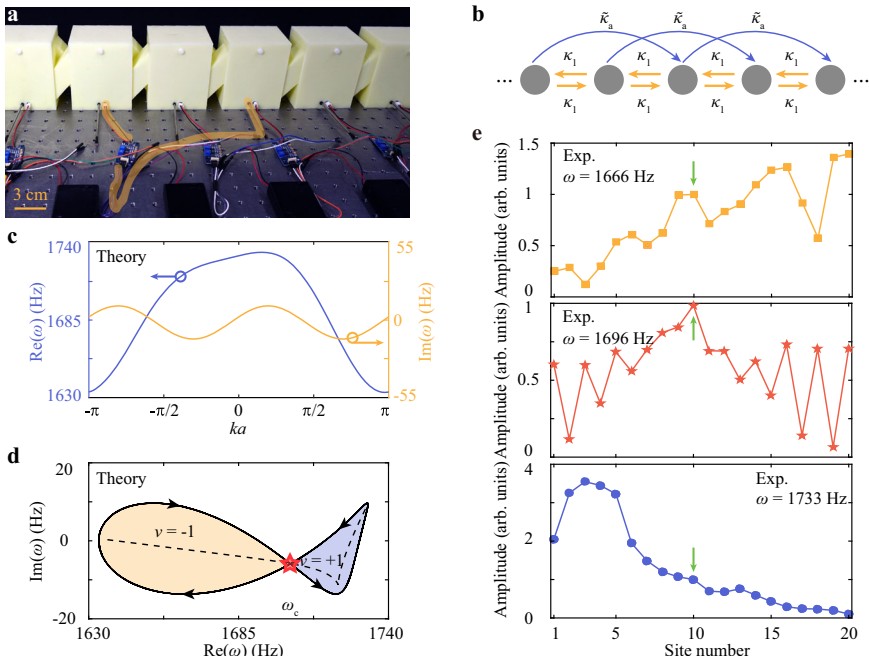

**Fig. 4 Observation of acoustic NHSE from twisted-winding topology. a** Photograph of the experimental setup, where the acoustic amplifiers connect the next-nearest-neighbor resonators. The yellow curve highlights an amplifier providing the non-reciprocal next-nearest-neighbor coupling. **b** Tight-binding model with reciprocal nearest-neighbor coupling $\kappa_1$ and non-reciprocal next-nearest-neighbor coupling $\tilde{\kappa}_a$. **c** Real and imaginary parts of the eigenfrequency spectrum calculated with periodic boundary conditions. **d** The complex eigenfrequency spectrum calculated in (**c**) forms a twisted winding with two opposite oriented loops in contact. The left loop takes the winding number $v = -1$, while the right one takes $v = +1$. The contact point, denoted as a red star, is the Bloch point. The dashed line corresponds to the eigenfrequency spectrum calculated with the open boundary conditions. **e** Measured field distributions when the source is fixed at site 10, but the frequency increases from $\omega = 1666, 1696, 1733$ Hz. The measured results are normalized by the amplitude at the source location (indicated by little arrows).

complex topological winding of a non-Hermitian band. Here we apply the non-reciprocal coupling $\tilde{\kappa}_a$ to the next-nearest-neighbor coupling, as shown in Fig. 4a, b. The nearest-neighbor coupling remains as $\kappa_1$, which is reciprocal. The corresponding lattice Hamiltonian with periodic boundary conditions can be written as $H = \kappa_1 e^{ika} + \kappa_1 e^{-ika} + \tilde{\kappa}_a e^{-2ika} + \omega_0 - i\gamma_0$. The imaginary and real parts of the eigenfrequency spectrum are presented in Fig. 4c. The corresponding eigenfrequency spectrum in the complex frequency plane is plotted in Fig. 4d. This time the eigenfrequency spectrum does not wind along a fixed direction but instead forms a twisted topology that consists of two opposite oriented loops in contact. The loop on the left-hand side winds in the clockwise direction, indicating a negative winding number $v = -1$. On the contrary, the loop on the right-hand side winds in the counter-clockwise direction, indicating a positive winding number $v = +1$. Moreover, the twisted winding inevitably features a transition point, named the Bloch point, at $\omega_c$, between the two loops, as highlighted with the red star in Fig. 4d. According to previous predictions, such a twisted winding will lead to bipolar localization, where the eigenstates collapse to two directions simultaneously[22]. The Bloch point will represent a Bloch-wave-like extended state that interpolates the left-localized and right-localized states[22].

Next, we experimentally measure the field distributions in the acoustic crystal exhibiting this twisted winding topology. We place a source at site 10 and measure the response at each site (see Methods). As depicted in Fig. 4e, at the frequency $\omega = 1666$ Hz, the wave excited from site 10 is strongly suppressed towards the left boundary but is dramatically amplified towards the right boundary, implying the winding number of $v = -1$. On the contrary, the phenomenon is reversed at the frequency $\omega = 1733$ Hz, indicating the flip of the sign of the winding number (see more results at different frequencies in

Supplementary Information Note 10). Notably, at a frequency $\omega = 1696$ Hz, which is between 1666 and 1733 Hz, the wave propagates towards both directions with almost identical amplitudes, with the maximum around the source. This is a manifestation of the Bloch-wave-like extended mode at the Bloch point[22]. The above experimental results match well with those from theoretical calculations (see Supplementary Information Note 11). Besides, we have measured the transmission of a pulse in the time domain (see Supplementary Information Note 12), and have experimentally investigated the pressure field distributions at different site numbers (see Supplementary Information Note 13). Taken together, we have thus experimentally demonstrated the twisted winding topology.

## Discussion

To conclude, we have demonstrated the NHSE in an active non-reciprocal acoustic crystal that is able to exhibit different winding topologies. By controlling the non-reciprocal coupling between arbitrary two sites, more complex unconventional topological windings can be achieved in the present acoustic platform. As the NHSE is a manifestation of the breakdown of the conventional bulk-boundary correspondence, it will be interesting to investigate the generalization of bulk-boundary correspondence and generalized Brillouin zones in non-Hermitian topological acoustics[14]. Though our experiments are carried out in 1D, it is feasible to extend the current design to two or three dimensions, greatly enriching the unique topology in non-Hermitian systems and enabling exotic physical phenomena such as high-order NHSE[28]. Furthermore, the feedback mechanism[29,30] is universal and can be applied to design arbitrary lattices with non-Hermitian, time-dependent, and even nonlinear coupling, in acoustics and electromagnetics. In terms of applications, the demonstrated

acoustic NHSE paves a way towards highly sensitive acoustic sensors, robust compact one-way waveguides, and amplifiers.

## Methods

**Experimental samples**. The acoustic samples in this work are fabricated with the 3D-printing technique with a fabrication error ~2 mm using photosensitive resin, which can be considered as acoustically rigid for airborne sound. The parameters of each acoustic resonator are height $h = 11.2$ cm, width $w = 7.2$ cm, length $l = 9.2$ cm, the distance between two acoustic resonators $s = 2.4$ cm, the two cross-linked narrow waveguides diameter $d = 3.4$ cm, and the thickness of the photosensitive resin is 6 mm, as illustrated in Figs. 2a, 3a, and 4a. Each active component comprises a directional amplifier (LM386 Low Voltage Audio Power Amplifier) equipped with a DC power supply, a speaker (output), and a microphone (input). Two holes (radius 4 mm) are drilled at the top on both sides of each acoustic resonator to insert the source and detector, respectively. The active component can provide the complex unidirectional coupling (see Supplementary Information Note 3). Two holes (radius 4 mm and 6.1 mm, respectively) are drilled at the bottom in the front of each acoustic resonator. The speaker (microphone) of the active component is inserted from the small (large) hole. The holes are sealed when not in use. The sample in Fig. 2a is composed of two cross-linked acoustic resonators and one active component. The sample in Fig. 3a is composed of 20 cross-linked acoustic resonators and 19 active components connecting two nearest-neighbor cavities. The sample in Fig. (4) is composed of 20 cross-linked acoustic resonators and 18 active amplifier components connecting two next-nearest neighbor cavities. Each active component is calibrated based on the two-resonator model.

**Numerical simulations**. The pressure field distribution of the acoustic resonator (Fig. 2c) is numerically calculated in the eigenfrequency of the commercial software COMSOL Multiphysics. The simulated model is the same as in Fig. 2a. The property of the materials is set as the air density $\rho_0 = 1.29$ kg m$^{-3}$ and the sound speed $c = 340 \times (1 + 0.0014i)$ m s$^{-1}$.

**Measurements**. A broadband sound signal is launched from a soft tube generated from the output generator module (B&K Type 3160-A-022), behaving like a point source, placed in the hole at the backside of the sample. The amplitude at each site is measured with a detector microphone (B&K Type 4182) placed in the hole at the front side of the sample, recorded with the signal acquisition module (B&K Type 3160-A-022). For the experiments in Fig. 2d and e, we measure the $S_{12}$ and $S_{21}$ by changing the position of the source and detector. The transmission spectra are normalized to the case when the source and the detector are directly connected. For the experiments in Figs. 3e, f, S5, S6, and S11, we place the source at sites 1 and 20, injecting a pulse ranging from 1680 to 1720 Hz, and measure the response pulse from site 20 and site 1, respectively. For the experiments in Figs. 3g, 4e, S7, S9, and S12, we measure the amplitude by changing the position of the detector at each site and polt the amplitude versus detector location.

## Data availability

The data that support the findings of this study can be obtained from the Zenodo database at https://doi.org/10.5281/zenodo.5564799.

## Code availability

The codes for numerical simulations are available in the Zenodo database at https://doi.org/10.5281/zenodo.5564799.

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

## Acknowledgements

The work at Zhejiang University was sponsored by the National Natural Science Foundation of China (NNSFC) under Grant nos. 61625502, 61975176, 11961141010, and 62175215, the Top-Notch Young Talents Program of China, and the Fundamental Research Funds for the Central Universities. The work at Jiangsu University was sponsored by the National Natural Science Foundation of China under Grant nos. 11774137, 12174159, and 51779107, and the State Key Laboratory of Acoustics, Chinese Academy of Science under Grant No. SKLA202016. Work at Nanyang Technological University was sponsored by Singapore Ministry of Education under Grant nos. MOE2019-T2-2-085 and MOE2016-T3-1-006.

## Author contributions

Y.Y., L.Z., B.Z., and H.C. conceived the original idea. L.Z., Y.Y., Y.G., and H.-x.S. designed the structures and the experiments. Y.G.Y.-J.G., D.J, and H.-X.S. conducted the experiments. L.Z., Y.Y., Q.C, Q.Y., R.X., Y.L., B.Z., and F.C. did the theoretical analysis. L.Z., Y.Y., B.Z., S.-Q.Y., and H.C. wrote the paper and interpreted the results. Y.Y., B.Z., H.-X.S., and H.C. supervised the project. All authors participated in discussions and reviewed the paper.

## Competing interests

The authors declare no competing interests.
