## [Peer Review File · Nature Communications]

Acoustic non-Hermitian skin effect from twisted winding topologyREVIEWER COMMENTS

Reviewer #1 (Remarks to the Author):

The manuscript presents an interesting experimental approach to Non Hermitian Skin Effect for 2 different winding topologies: single and twisted. Near neighbor non-reciprocal coupling interaction provides the single winding topology, while the next near neighbor NR interaction provides the twisted one.

I have a few questions:

- 1) Why there a need for two rather than one reciprocal coupling between the resonators (two narrow wave-guides)?
- 2) For the dimmer experiments in Fig 2e, why is the amplitude S_{12} smaller than S_{21} . Is there a difference on how S_{12} and S_{21} measured?
- 3) Fig 3e the source is at one end and detector at the other end. At what site number the no transmission to the right breaks? In other words what is the acoustic profiles source is at site 1 and measurements are site 2,3, ..20.
- 4) Fig 3e, how much is the increase in the amplitude and how is this compared with how much energy is pumped into the system.
- 5) Fig 3g presents only one frequency. Has the remaining of the frequencies been measured? Is the profile similar?
- 6) Ideally, I would like to see the experiments done in Figure 3e also for next near neighbor system.
- 7) Fig 4e Ideally I would like to see the profile for more frequencies for both winding numbers.
- 8) Fig 4e: have you seen and increase in amplitude at one end, as the excitation is moved at higher site numbers?

Reviewer #2 (Remarks to the Author):

The authors demonstrate non Hermitian skin effect in an active non-reciprocal acoustic crystal, being able exhibit different winding topology. This is achieved by controlling the non-reciprocal coupling between the sites of a metamaterial array. I found the study kind of interesting and timely. The authors

have fully demonstrated the concept by providing rigorous experimental analysis. The manuscript is written well and organized nicely. Given these important factors, I think the manuscript can be published in Nature Communications after addressing the following minor comments:

1- The equations are not labeled in the manuscript.

2- The scale bar in Fig. 1a and 3a are missing.

3- It is not clear to me how the proposed structure corresponds to a simple tight binding model discussed by the authors. Could the authors clarify on this a bit more?

4- There is no direct cross validation of the experimental result using theoretical and or simulations ones. In particular, it would be great if the results of Fig. 4e were compared with simulations.

5- The authors have not considered losses in their simulations. In particular they have modeled air with a material with a purely real density. This is not the case in reality. In order to be able achieve simulation results comparable to the experimental results, one should also include some visco-thermal losses in simulation. I would recommend the authors to revisit this.

Response Letter to Reviewers

We are grateful for the useful comments on this manuscript (NCOMMS-21-15646-T) from all the reviewers.

In the text below each of the comments from each reviewer is quoted in *Italic* and is followed by the corresponding detailed response. We also revised the manuscript and the Supplementary Information accordingly, and these updates are highlighted in blue or by a vertical red line in the left margin in those files. In the text below the references to these updates are highlighted in a similar way (i.e., by a vertical red line).

GENERAL COMMENTS FROM 1st REVIEWER:

The manuscript presents an interesting experimental approach to Non Hermitian Skin Effect for 2 different winding topologies: single and twisted. Near neighbor non-reciprocal coupling interaction provides the single winding topology, while the next near neighbor NR interaction provides the twisted one.

Response from Authors:

We thank the reviewer for the encouraging and favourable comments. In particular, we are glad that the reviewer considers our work “*presents an interesting experimental approach*”. In the following, we address the specific comments point-by-point.

SPECIFIC COMMENTS FROM 1st REVIEWER:

1st Reviewer -- Comment 1:

1. Why there a need for two rather than one reciprocal coupling between the resonators (two narrow waveguides)?

Response from Authors:

The reason why we use two waveguides for the reciprocal coupling is to enhance the coupling strength. This can help to obtain a relatively wide operating frequency range.

We have added a note on page 4 of the main text, starting from line 98, which reads as,

“**Note that the two waveguides can enhance the coupling strength.**”

1st Reviewer -- Comment 2:

2. For the dimmer experiments in Fig 2e, why is the amplitude S_{12} smaller than S_{21} . Is there a difference on how S_{12} and S_{21} measured?

Response from Authors:

The S_{12} and S_{21} are measured in the same way except for the swapped positions of the source and detector. The difference between their amplitudes is due to the non-reciprocity arising from the unidirectional acoustic amplifier. Without the amplifier, the S_{12} and S_{21} have the same amplitude (see Fig. 2d).

1st Reviewer -- Comment 3:

3. Fig 3e the source is at one end and detector at the other end. At what site number the no transmission to the right breaks? In other words what is the acoustic profiles source is at site 1 and measurements are site 2,3, ..20.

Response from Authors:

Following the reviewer's suggestion, we have measured the corresponding acoustic profiles at different sites, as shown in Fig. R1 below. It can be seen that the transmission toward the right boundary (site 20) decreases dramatically as the site number increases. Around site 17, the transmission is almost zero.

We have added a discussion on page 7 of the main text, starting from line 153, which reads as, "As depicted in Figs. 3e and f, the transmission towards the right boundary almost vanishes, while the transmission towards the left boundary is amplified (also see the measured time-domain results in Supplementary Information Note 6 and Note 7)."

Besides, we have included Fig. R1 as Supplementary Fig. S5 in a new section of Supplementary Information entitled "Supplementary Note 6: Transmission measurement of a pulse in the time domain in the acoustic crystal with the nearest-neighbor non-reciprocal coupling".

Figure R1. Transmission measurement of a pulse in the time domain from site 1 to site 20 in the acoustic crystal with the nearest-neighbor non-reciprocal coupling. The measured results are normalized by the amplitude at site 1.

1st Reviewer -- Comment 4:

4. Fig 3e, how much is the increase in the amplitude and how is this compared with how much energy is pumped into the system.

Response from Authors:

In Fig. 3e, a Gaussian-like pulse, covering the frequency range from 1680 to 1720 Hz, is launched from site 1 and detected at site 20. The transmission towards the right boundary almost vanishes. In order to compare the decrease in the amplitude, we apply the Fourier transform to obtain the corresponding frequency spectra [Fig. R2(a) and R2(b)]. The decrease in the amplitude is defined as the difference between the amplitude of the input pulse and that of the detected pulse, which is shown in Fig. R2(c).

In addition, the decay ratio defined as the ratio between the decrease in the amplitude and amplitude of the input pulse is also plotted in Fig. R2(d).

In the case of Fig. 3e, the energy is not pumped into, but is pumped out (almost 100%) of, the system.

We have added a discussion on page 7 of the main text, starting from line 153, which reads as, “As depicted in Figs. 3e and f, the transmission towards the right boundary almost vanishes, while the transmission towards the left boundary is amplified (also see the measured time-domain results in Supplementary Information Note 6 and Note 7).”

We have also added a new section of Supplementary Information entitled “Supplementary Note 7: Decrease in amplitude and decay ratio”. Figure R2 is included as Supplementary Fig. 6. In this Supplementary Note 7, we have explained on page 7, in line 115, that “In this case, the energy is pumped out of the system.”

Figure R2. Decrease in the amplitude and the decay ratio of a pulse in the time domain. a, b, Frequency spectra obtained by applying Fourier transform to the time-domain signal at site 1 and site 20. The measured results are normalized by the amplitude at site 1. **c,** Measured decrease in amplitude as a function of frequency. The measured results are normalized by the amplitude at site 1. **d,** Measured decay

ratio as a function of frequency. The blue region represents three standard deviations of the Gaussian-like pulse, which covers a frequency range from 1680 to 1720 Hz.

1st Reviewer – Comment 5:

5. Fig 3g presents only one frequency. Has the remaining of the frequencies been measured? Is the profile similar?

Response from Authors:

In Fig. R3, we show the measured field profiles at different frequencies, which are very similar to those in Fig. 3g. When the source is placed further away from the left, the field distribution localized at the left boundary exhibits stronger amplitude.

We have added a discussion on page 7 of the main text, starting from line 157, which reads as, “Moreover, we also measure the field distributions in the acoustic crystal when the source is placed at different locations with operating frequency $\omega = 1713$ Hz, as shown in Fig. 3g (see more results at other frequencies in Supplementary Information Note 8).”

Besides, we have also included Fig. R3 in a new section of Supplementary Information entitled “Supplementary Note 8: Measured field intensity distributions in the acoustic crystal with the nearest-neighbor non-reciprocal coupling”.

Figure R3. Measured field intensity distributions in the acoustic crystal with the nearest-neighbor non-reciprocal coupling. a-c, Field intensity distribution at frequency $\omega = 1684$, 1704 , and 1724 Hz, when the source is located at site 6, 8, and 10, respectively. The measured results are normalized by the amplitude at the source location (indicated by little arrows).

1st Reviewer -- Comment 6:

6. Ideally, I would like to see the experiments done in Figure 3e also for next near neighbor system.

Response from Authors:

Following the reviewer's suggestion, we have measured the transmission of a pulse, covering the frequency range from 1600 to 1800 Hz, in the time domain for the sample with non-reciprocal next-nearest-neighbor couplings, as shown in Figs. R4(a) and (c).

We also apply the Fourier transform to the signal to obtain the corresponding frequency spectra. Due to the twisted winding topology, the pulse is amplified towards the *right* boundary within frequency ranging from 1647 to 1675 Hz [see Fig. R4(b)]. On the contrary, the phenomenon is reversed when the pulse is launched at site 20, where the pulse is amplified towards the *left* boundary within frequency ranging from 1714 to 1736 Hz [see Fig. R4(d)].

We have also added a discussion on page 10 of the main text, starting from line 211, which reads as,

“Besides, we have measured the transmission of a pulse in the time domain (see Supplementary Information Note 12)”.

Besides, we have included Fig. R4 in a new section of Supplementary Information entitled “Supplementary Note 12: Transmission measurement of a pulse in the time domain in the acoustic crystal with the next-nearest-neighbor non-reciprocal coupling”.

Figure R4. Transmission measurement of a pulse in the time domain in the acoustic crystal with the next-nearest-neighbor non-reciprocal coupling. **a, c**, Transmission measurement of a pulse launched at site 1 (20) in the time domain with frequencies ranging from 1600 to 1800 Hz, respectively. The measured results are normalized by the amplitude at the source location. **b, d**, Frequency spectra obtained by applying Fourier transform to the time-domain signal. The measured results are normalized by the amplitude at the source location. The top panel in each figure shows the schematic diagram with the source indicated by a hexagon and the detector marked by a star.

1st Reviewer -- Comment 7:

7. Fig 4e Ideally I would like to see the profile for more frequencies for both winding numbers.

Response from Authors:

Following the reviewer's suggestion, we have plotted Fig. R5 to show the measured field distributions at more frequencies for both winding numbers. One can see that below the frequency of the Bloch point (around 1696 Hz), the wave excited from site 10 is strongly suppressed towards the left boundary, but is dramatically amplified towards the right boundary, implying the winding number of $\nu = -1$. On the contrary, the phenomenon is reversed above the Bloch point frequency, indicating the flip of the sign of the winding number.

We have also added a discussion on page 9 of the main text, starting from line 205, which reads as,

“On the contrary, the phenomenon is reversed at the frequency $\omega = 1733$ Hz, indicating the flip of the sign of the winding number (see more results at different frequencies in Supplementary Information Note 10).”

Besides, we have included Fig. R5 in a new section of Supplementary Information entitled

“Supplementary Note 10: Measured field intensity distributions in the acoustic crystal with the next-nearest-neighbor non-reciprocal coupling”.

Figure R5. Measured field intensity distributions in the acoustic crystal with the next-nearest-neighbor non-reciprocal coupling. a-h, Field intensity distributions at different frequencies. The measured results are normalized by the amplitude at the source location (indicated by little arrows).

1st Reviewer -- Comment 8:

8. Fig 4e: have you seen and increase in amplitude at one end, as the excitation is moved at higher site numbers?

Response from Authors:

Thanks for the constructive suggestion.

Following the reviewer's suggestion, we have done additional measurements. Below the Bloch point frequency (around 1690 Hz), the wave intensity at the *right* boundary increases as the excitation moves to the *lower* site number [see Fig. R6(a)]. Above the Bloch point frequency, the wave intensity at the *left* boundary increases as the excitation moves to the *higher* site number [see Fig. R6(c)]. At the Bloch point, the wave intensity at both boundaries is insensitive to the excitation position [see Fig. R6(b)].

We have also added a discussion on page 10 of the main text, starting from line 213, which reads as,

“and have experimentally investigated the pressure field distributions at different site numbers (see Supplementary Information Note 13).”

Besides, we have included Fig. R6 in a new section of Supplementary Information entitled “Supplementary Note 13: Measured field intensity distributions with different excitations in the acoustic crystal with the next-nearest-neighbor non-reciprocal coupling”.

Figure R6. Measured field intensity distributions in the acoustic crystal with the next-nearest-neighbor non-reciprocal coupling. a-c, Measured field intensity distribution at frequency $\omega = 1660, 1690,$ and 1727 Hz, when the source is located at the site 6, 8, and 10, respectively. The measured results are normalized by the amplitude at the source location (indicated by little arrows).

GENERAL COMMENTS FROM 2nd REVIEWER:

The authors demonstrate non Hermitian skin effect in an active non-reciprocal acoustic crystal, being able exhibit different winding topology. This is achieved by controlling the non-reciprocal coupling between the sites of a metamaterial array. I found the study kind of interesting and timely. The authors have fully demonstrated the concept by providing rigorous experimental analysis. The manuscript is written well and organized nicely. Given these important factors, I think the manuscript can be published in Nature Communications after addressing the following minor comments:

Response from Authors:

We thank the reviewer for the encouraging comments and recommendation for publication in *Nature Communications*. In particular, we are glad that the reviewer considers “*the study kind of interesting and timely. The authors have fully demonstrated the concept by providing rigorous experimental analysis. The manuscript is written well and organized nicely.*” In the following, we fully address the specific comments point-by-point.

SPECIFIC COMMENTS FROM 2nd REVIEWER:

2nd Reviewer -- Comment 1:

1. *The equations are not labeled in the manuscript.*

Response from Authors:

Following the reviewer's suggestion, we have labelled the equations in our revised manuscript.

2nd Reviewer -- Comment 2:

2. *The scale bar in Fig. 1a and 3a are missing.*

Response from Authors:

Following the reviewer's suggestion, we have added the scale bar in Figure 2a, 3a, and 4a, respectively.

2nd Reviewer -- Comment 3:

3. *It is not clear to me how the proposed structure corresponds to a simple tight binding model discussed by the authors. Could the authors clarify on this a bit more?*

Response from Authors:

We appreciate the reviewer for his/her suggestions.

Due to the versatile tunability and ease of fabrication, the acoustic system is widely adapted for realizing tight-binding models. Here, we show an example to briefly introduce the connection between the designed acoustic structure and the tight-binding model.

As depicted in Fig. R7(a), a single acoustic resonator has a dipole mode with eigenfrequency $\omega_0 = 1700$ Hz. When two resonators are connected by narrow waveguides, the coupling between two dipole modes will lead to eigenfrequency splitting. The resulting symmetric (at 1685 Hz) and antisymmetric (at 1728 Hz) modes are shown in Fig. R7(b). Such a two-resonator system can be

described by a two-level Hamiltonian $H = \begin{bmatrix} \omega_0 & \kappa \\ \kappa & \omega_0 \end{bmatrix}$, with κ being the coupling strength.

Continuously increasing the number of resonators, the eigenfrequency splitting process will result in many eigenfrequencies covering a certain frequency range. Eventually, the coupled resonators form a 1D infinite-size periodic chain that has a continuous band structure, which can be described by a Hamiltonian $H = \kappa e^{ika} + \kappa e^{-ika} + \omega_0$, with k being the wavevector and a being the lattice constant [see Fig. R7(c)].

We have added a new section of Supplementary Information entitled

“Supplementary Note 14: Tight-binding model for the coupled acoustic resonators”.

Figure R7. Tight-binding model for the coupled acoustic resonators. **a**, Eigen frequency and eigen mode of a single acoustic resonator. **b**, Eigen frequencies and eigen modes of two coupled resonators. **c**, Band structure and an eigenmode of a 1D chain consisting of coupled resonators.

2nd Reviewer -- Comment 4:

4. There is no direct cross validation of the experimental result using theoretical and or simulations ones. In particular, it would be great if the results of Fig. 4e were compared with simulations.

Response from Authors:

Following the reviewer's suggestion, we have done theoretical calculations to compare them with the experimental results.

According to the coupled-mode theory, when the wave is incident to site 10 and detected at site 20, the dynamic equation can be described as,

$$\begin{aligned}
 da_1/dt &= (-i\omega_0 - \gamma_0)a_1 - i\kappa_1 a_2 \\
 da_2/dt &= -i\kappa_1 a_1 + (-i\omega_0 - \gamma_0)a_2 - i\kappa_1 a_3 \\
 da_3/dt &= -i\tilde{\kappa}_a a_1 - i\kappa_1 a_2 + (-i\omega_0 - \gamma_0)a_3 - i\kappa_1 a_4 \\
 &\dots \\
 da_{10}/dt &= -i\tilde{\kappa}_a a_8 - i\kappa_1 a_9 + (-i\omega_0 - \gamma_0 - \gamma_1)a_{10} - i\kappa_1 a_{11} + \sqrt{2\gamma_1} s_{1+} \\
 &\dots \\
 da_{19}/dt &= -i\tilde{\kappa}_a a_{17} - i\kappa_1 a_{18} + (-i\omega_0 - \gamma_0)a_{19} - i\kappa_1 a_{20} \\
 da_{20}/dt &= -i\tilde{\kappa}_a a_{18} - i\kappa_1 a_{19} + (-i\omega_0 - \gamma_0 - \gamma_1)a_{20} \\
 s_{2-} &= \sqrt{2\gamma_1} a_{20} \\
 s_{21} &= s_{2-} / s_{1+} = \sqrt{2\gamma_1} a_{20} / s_{1+}
 \end{aligned}$$

where a_n is the mode in site n , s_{1+} represents the incident wave from site 10, s_{2-} represents the detected output wave from site 20, and s_{21} represents the transmission coefficient. Similarly, the transmission can also be obtained when the wave is detected at each site. Then we plot the transmission versus the site number at different operating frequencies. As shown in Fig. R8, the theoretical results match well with the experimental counterparts (see Fig. 4e).

We have added a discussion on page 10 of the main text, starting from line 210, which reads as,

“The above experimental results match well with those from theoretical calculations (see Supplementary Information Note 11).”

We have also added a new section of Supplementary Information entitled “Supplementary Note 11: Theoretical calculation of field intensity distributions in the acoustic crystal with the next-nearest-neighbor non-reciprocal coupling”.

Figure R8. Calculated field intensity distributions in the acoustic crystal with the next-nearest-neighbor non-reciprocal coupling. a-c, Calculated field intensity distributions in the acoustic crystal with the next-nearest-neighbor non-reciprocal coupling, when the source is placed at site 10 at frequency $\omega = 1666$, 1699 , and 1727 Hz, respectively. The measured results are normalized by the amplitude at site 10.

2nd Reviewer -- Comment 5:

5. The authors have not considered losses in their simulations. In particular they have modeled air with a material with a purely real density. This is not the case in reality. In order to be able achieve simulation results comparable to the experimental results, one should also include some visco-thermal losses in simulation. I would recommend the authors to revisit this.

Response from Authors:

We thank the reviewer for this constructive suggestion.

The intrinsic loss, including the viscothermal loss, has been obtained by fitting the experimental results to the coupled-mode model. The resulting intrinsic loss is $\gamma_0 = 2.13$ Hz in our experiments. We should note that the intrinsic loss only slightly shifts the band structure in the complex frequency plane and does not change the non-Hermitian topology. Thus, the main conclusions in our work remain valid.

To include the intrinsic loss in our simulation, we set the sound speed as $340 \times (1 + 0.0014i)$ m s⁻¹, which matches with our experimental results very well.

Besides, to include the influence of the losses, we have added an extra loss term in the Hamiltonians, i.e., $H=\kappa e^{jka} + \kappa e^{-jka} + \tilde{\kappa}_a e^{-ika} + \omega_0 - i\gamma_0$ for the single-winding topology and $H=\kappa e^{jka} + \kappa e^{-ika} + \tilde{\kappa}_a e^{-2ika} + \omega_0 - i\gamma_0$ for the twisted-winding topology.

We have added a note on page 4 of the main text, starting from line 103, which reads as,

“The tight-binding model (see Fig. 2b) can lead to the following Hamiltonian (see details in Supplementary Information Note 3):

$$H = \begin{bmatrix} \omega_0 - i\gamma_0 & \kappa_1 \\ \kappa_1 + \tilde{\kappa}_a & \omega_0 - i\gamma_0 \end{bmatrix}. \quad (1)$$

Here, γ_0 arises from the intrinsic loss, including the viscothermal loss.”

We have modified the Hamiltonian on page 6 of the main text, starting from line 135, which reads as,

“The lattice Hamiltonian of the crystal with periodic boundary conditions is $H=\kappa e^{jka} + \kappa e^{-ika} + \tilde{\kappa}_a e^{-ika} + \omega_0 - i\gamma_0$.”

We have also modified the Hamiltonian on page 9 of the main text, starting from line 187, which reads as,

“The corresponding lattice Hamiltonian with periodic boundary conditions can be written as $H=\kappa e^{jka} + \kappa e^{-ika} + \tilde{\kappa}_a e^{-2ika} + \omega_0 - i\gamma_0$.”

We have updated Figures 3, 4, and S1.

We have added a note in Methods on page 12 of the main text, starting from line 265, which now reads as,

“The property of the materials is set as the air density $\rho_0= 1.29 \text{ kg m}^{-3}$ and the sound speed $c = 340 \times(1+0.0014i) \text{ m s}^{-1}$.”

REVIEWERS' COMMENTS

Reviewer #1 (Remarks to the Author):

The authors answered all my questions. I recommend publication.

Reviewer #2 (Remarks to the Author):

The authors have properly addressed my previous comments, so I recommend the publication of the work at this stage.

Response Letter to Reviewers

We are grateful for the useful comments on this manuscript (NCOMMS-21-15646A) from all the reviewers.

In the text below each of the comments from each reviewer is quoted in *Italic* and is followed by the corresponding detailed response.

GENERAL COMMENTS FROM 1st REVIEWER:

The authors answered all my questions. I recommend publication.

Response from Authors:

We thank the reviewer for the favorable recommendation.

GENERAL COMMENTS FROM 2nd REVIEWER:

The authors have properly addressed my previous comments, so I recommend the publication of the work at this stage.

Response from Authors:

We thank the reviewer for the recommendation for publication in *Nature Communications*.